# Rise of Social Media Influencers as a New Marketing Channel: Focusing on the Roles of Psychological Well-Being and Perceived Social Responsibility among Consumers

**DOI:** 10.3390/ijerph19042362

**Published:** 2022-02-18

**Authors:** Jihye Kim, Minseong Kim

**Affiliations:** 1Department of Integrated Strategic Communication, College of Communication and Information, University of Kentucky, Lexington, KY 40506, USA; jihye.kim@uky.edu; 2Department of Management & Marketing, College of Business, Louisiana State University Shreveport, Shreveport, LA 71115, USA

**Keywords:** social media, social media influencer, digital attributes, psychological well-being, friendship, loyalty, social responsibility, parasocial interaction

## Abstract

This empirical research investigated the structural relationships between social media influencer attributes, perceived friendship, psychological well-being, loyalty, and perceived social responsibility of influencers, focusing on the perspective of social media users. More specifically, this study conceptually identified social media influencer attributes such as language similarity, interest similarity, interaction frequency, and self-disclosure and examined the respective effects of each dimension on perceived friendship and psychological well-being, consequently resulting in loyalty toward social media influencers. The authors collected and analyzed data from 388 social media users in the United States via Amazon’s Mechanical Turk with multivariate analyses to test the hypothesized associations among the variables in this study. The findings indicated that perceived friendship was significantly influenced by language similarity, interest similarity, and self-disclosure, but did not have a significant impact on psychological well-being. Additionally, perceived friendship significantly affected psychological well-being and loyalty, and psychological well-being significantly influenced loyalty. Lastly, social media influencers’ social responsibility moderated the path from psychological well-being to loyalty. Based on these findings, this study proposes theoretical and managerial implications for the social media influencer marketing context.

## 1. Introduction

Social media use is widespread and has been shown to play an important role in the lives of both adolescents and young adults [1]. Social media platforms enable adolescents and young adults to consume digital content without time and space limits and to virtually interact with other users without geographical boundaries [2]. Particularly, the development of digital platforms leads young adults to closely interact with their favorite celebrities, brands, and other users in the virtual world via two-way communication tools, such as live chats and comment options [3]. Interestingly, as social media platforms have grown, the digital world created a new phrase, “social media influencers,” who become famous through their digital content on social media, compared to traditional celebrities who become famous on TV shows and film [4]. Hence, social media users tend to feel connected with social media influencers by interacting with them in the virtual world and perceive social media influencers as more authentic in their fields, including fashion, health, or music, than celebrity endorsements in traditional advertisements [1]. Therefore, there is a need to empirically investigate what drives social media users’ loyalty toward their favorite influencers, such as repeat purchase behavior, positive word-of-mouth, and recommendation of products advertised by social media influencers from the perspectives of an influencer’s digitalized attributes [5,6]. These perspectives are especially important because social media users interact with their favorite influencers in the digital world instead of in the real world. This means that social media influencers’ digitalized attributes can be more important determinants of users’ perceptions, feelings, behavioral intention, and even actual behaviors than the visual characteristics of social media influencers. However, prior research in this field focused primarily on the visual and/or actual characteristics of social media influencers (e.g., physical attractiveness or social attractiveness) to predict marketing outcomes, such as positive attitudes toward and favorable intention for a product advertised by the influencers [7]. In particular, loyalty toward social media influencers leads followers to have a strong sense of product/brand association and reality by perceiving influencers’ product/brand and advertisements as more persuasive and authentic [4]. Accordingly, users are more likely to behave favorably for the product/brand social media influencers advertise in order to endorse the influencers and support the influencers’ fame and social status [7].

Specifically, this study considers psychological well-being as one of the core determinants of social media users’ loyalty toward their favorite influencers by focusing on the fundamental motivations of consuming digital content among users, such as enjoyment, pleasure, happiness, and friendship [6]. For example, according to Kim and Kim [3], social media platforms provide users with virtual spaces to consume celebrities’ digital content and interact with social media influencers and other users in the social media community, leading users to form positive emotions in the digital world that transfer to happiness and pleasure in the real world. Accordingly, this study draws from the parasocial interaction, the self-congruity, and the psychological well-being theories to propose that social media users develop perceived friendships with their favorite influencers in the virtual world by consuming the influencers’ digital content and interacting with the influencers, resulting in the development of psychological well-being among users in the real world [7]. More specifically, the digitalized interactive features of social media platforms enable users to feel emotionally tied to their favorite social media influencers (i.e., users perceive the influencers as intimate friends in the real world via virtual interactions), including social foci, proximity, interaction frequency, and self-disclosure [3,8]. Through the social media features, users tend to feel like they develop a solid friendship with their favorite social media influencers and to be satisfied with their real life [7,8]. Therefore, based on the parasocial interaction theory and prior studies on social media [3,7,9], this study identifies language similarity, interest similarity, interaction frequency, and self-disclosure as social media influencer attributes that serve as core determinants of users’ perceived friendship with their favorite social media influencers and psychological well-being in real life.

Importantly, in the social media influencer marketing context, consumers develop their own beliefs regarding why their favorite influencers promote and advertise a product/brand and whether the influencers as well as the product/brand are socially acceptable or not. Social media users may be skeptical about socially unacceptable brands even though their favorite influencers have a strong partnership with the brands, and vice versa [10]. Nowadays, media coverage has reported that social media influencers have created digital content without ethical standards, and media coverage has disclosed socially unacceptable scandals of social media influencers [10,11]. Social media influencers’ wrongdoing or deviance has created spillover effects on brands and/or products that have partnerships with the social media influencers [10]. Therefore, users’ perceived social responsibility of social media influencers needs to be studied as a moderator in the context of social media influencer marketing. Some brands have partnerships with social media influencers based on the influencers’ number of loyal followers without consideration of the influencers’ socially responsible aspects. Based on the notion of the spillover effect, this study assumes that the social values and socially responsible behaviors of social media influencers, such as money and time donations for people in need and support for nonprofit organizations, have significantly influenced their followers’ and other users’ perceptions about the influencers and digital content (i.e., including advertisements) [12]. To the best of our knowledge, although social media users’ perceived social responsibility has been well-studied from the perspective of brands or companies [13], no empirical research has been conducted from the perspective of users’ perceived social responsibility of social media influencers playing a role as brand ambassadors and advertising agencies in social media, such as a spokesperson.

In conclusion, this study investigates the distinct effects of social media influencer attributes on users’ perceived friendships with their favorite social media influencers and psychological well-being, consequently resulting in loyalty toward the influencers and advertised products. More importantly, this research examines the moderating role of users’ perceived social responsibility of their favorite social media influencers in the relationships between perceived friendship/psychological well-being and loyalty. From a theoretical perspective, this empirical research contributes to the existing literature on social media influencer marketing by extending the parasocial interaction theory considering the digitalized attributes of social media influencers compared to previous studies that emphasized influencers’ visual attributes [7]. Additionally, this research extends the parasocial interaction theory by considering the roles of the perceived social responsibility of social media influencers and psychological well-being among digital users, reflecting the current trend in the marketing field (i.e., the importance of social responsibility and psychological well-being in the era of COVID-19) [3,6]. From a managerial standpoint, this study proposes the important role of social media influencers’ social responsibility in formulating a partnership between a brand (or company) and the influencers.

## 2. Literature Review and Hypotheses Development

### 2.1. The Parasocial Interaction Theory as a Theoretical Backgroud

The parasocial interaction theory was developed from the perspectives of developing relationships between spectators and mass media, such as television and radio [14]. As an illusionary experience, the parasocial interaction is based on the notion that spectators perceive that media-mediated representations on mass media, such as characters, celebrities, or presenters, are talking directly to them, engaging in reciprocal relationships [15]. After perceiving it, spectators begin to consider media-mediated representators as their real friends according to the representators’ nonverbal and verbal interaction cues on mass media [16,17]. If the indirect interactions between spectators and representators continue, spectators’ feelings about friendships with the representators can be enduring and strengthened [18].

The parasocial interaction theory has been applied to computer-mediated environments that bring users closer to a computer-mediated person via communication tools [16]. More specifically, computer-mediated environments provide users with direct two-way communications between users and/or between a user and a celebrity/influencer, generating virtual interactions to enable the user to view the celebrity/influencer as a real friend [19]. Additionally, consistent with the components of traditional mass media, social media users use verbal and nonverbal cues of influencers to foster parasocial interactions [3]. More specifically, social media influencers may use digitalized verbal and nonverbal cues by demonstrating similar language styles and similar interests as their followers to facilitate a feeling of psychological proximity, leading the followers to develop a parasocial relationship with the influencers [19]. Additionally, social media users who directly and indirectly interact with their favorite influencers more frequently tend to report a strong feeling of parasocial relationships with the influencers. This is primarily because users feel like the interactions with the influencers on social media are ordinary meetings with real friends, according to the notion of the parasocial interaction theory [16]. Interestingly, as social media users are exposed to their favorite influencers’ personal stories over time, they form beliefs and opinions about the influencers by developing parasocial relationships with the influencers in the virtual world, similar to in the real world [9]. Therefore, this study proposes language similarity, interest similarity, interaction frequency, and self-disclosure as verbal and nonverbal cues of influencers (i.e., social media influencer attributes in this study) [3,9].

### 2.2. Social Media Influencer Attributes

In this study, similarities between social media users and influencers regarding personality, values, or beliefs are referred to as self-congruity based on the self-congruity theory that explains how social media users select their favorite influencers [3,9]. According to the self-congruity theory, social media users select and consume their favorite influencers’ digital content consistent with their self-concepts [20]. This study proposes that social media users’ perceived consistency with images of influencers is based on the sources of self-congruity, such as perceived similar language styles and messages the influencers use to communicate with social media followers and spectators [20,21]. For example, perceived language similarity can be formed by communication styles, phrases, and words between social media users and influencers, and perceived interest similarity can be formed by influencers’ digital content [21]. In particular, the social media influencers’ digital content can create more opportunities for disclosing personal information with other users if the digital content is more frequently uploaded and updated [3,22]. Thus, social media users have more opportunities to interact with their favorite influencers and learn more about their favorite influencers’ images over time [23]. Lastly, social media users’ perceived images of their favorite influencers can also be influenced by the influencers’ personal thoughts, information, and feelings intentionally shared with the users on social media [22,23]. The shared personal information results in users’ perceived intimacy with the influencers, strong connections and emotional exchanges [3,9].

### 2.3. Perceived Friendship

In consumer behavior, the concept of friendship has been considered to be a close relationship between service providers and consumers [24,25]. Additionally, consumers’ perceptions of friendship can be formed by a close relationship with intangible things, such as brands and digitalized content, via interpersonal interactions on social media [9,26]. However, the interpersonal interactions between users on social media should be based on inclusion, loyalty, reciprocity, mutual trust, and intimacy and care between the users [24]. Due to the components of interpersonal interactions, social media users develop and maintain friendships with other users by reducing limitations of the virtual world, such as uncertainty, intangibility, and complexity [3]. In addition, for interpersonal purposes, users can employ particular tools of social media platforms, including clicking the like and share buttons and leaving comments on others’ posts, leading users to feel like they are interacting with real friends in the virtual world [3,25,26]. Therefore, social media platform tools serve as a full mediator of building a perceived friendship between users in the virtual world [26].

### 2.4. Psychological Well-Being

In general, psychological well-being refers to life satisfaction and fulfillment, including individuals’ emotional reactions to both short moments and long periods of time [27,28]. The psychological well-being theory assumes that the short moments and long periods of time should fulfill individuals’ social, psychological, and physical needs to maintain a high level of psychological well-being [28,29,30,31]. Interestingly, the extant literature indicates the positive relationship between psychological well-being and the use of social media platforms, such as digital content consumption and virtual interactions with others on social media, by highlighting that it increases individuals’ feelings of belonging and provides individuals with confidence in life [27,28,30]. Therefore, social media platforms have the potential to meet users’ social, psychological, leisure, and even physical needs, increasing their level of psychological well-being [30]. However, heavy social media users may often displace social activities in the real world by being addicted to social media use, focusing on maintaining their life satisfaction primarily via digital activities [30]. In addition, although heavy social media users can maintain their psychological well-being, excessive digital activities may generate detrimental effects on users’ physical health, such as body composition and quality of sleep [29]. Thus, heavy digital media use may result in deleterious consequences for social media users.

### 2.5. Loyalty

On social media, user loyalty refers to influencers’ ability to keep users consuming digital content created by the influencers over other influencers [32,33]. Hence, social media user loyalty can be displayed as primarily consuming the influencers’ digital content, although there are many alternatives who create digital content similar to that of their favorite influencer [34]. Social media user loyalty also tends to be expressed as positive word-of-mouth and recommendation of the influencer to other users [33]. In other words, users who are loyal toward a particular influencer are more likely to play a role as an active advocator or supporter for the influencer in the real world as well as in the virtual world [34]. Interestingly, social media user loyalty toward favorite influencers consequently leads to users’ favorable behaviors for products/services the influencer advertises and promotes, including purchase behavior, positive word-of-mouth, and recommendation of the products/services to relatives, friends, and others on social media [33,35]. The users tend to commit a substantial amount of money and time to the products/services without reservation only because their favorite influencer advertises and promotes the products/services [35]. Therefore, it is especially important for social media influencers to build user loyalty by retaining their followers and spectators on a long-term basis, bringing the influencers reputation and profitability [32,33].

### 2.6. Research Hypotheses

According to the social media literature, users are more likely to develop and maintain strong relationships with others who have similar interests [3,5]. Within the virtual world, perceived similarity enables users to mitigate potential conflicts and misunderstandings of other users when building relationships on social media [36]. Additionally, social media users are more likely to communicate with others more frequently and effectively if they have the same language styles and interests in particular topics, including between a user and a user and between a user and a social media influencer [3,5,36]. More importantly, on social media platforms, users’ similarity with others can also be perceived by user-generated digital content that includes language and communication style in addition to physical similarity (e.g., location and gender) [9,37]. Furthermore, social media platforms provide influencers with enough space and storage to offer digital content to other users without time and space limits [2,37]. Thus, social media users can always consume their favorite influencers’ digital content and interact with them, leading them to develop friendships between users and influencers via frequent interactions [3]. Additionally, influencers who reveal personal interests, thoughts, and beliefs to users on social media may form an intimacy and emotional reliance [37]. Therefore, this study argues that digital content and attributes, including influencers’ characteristics, may lead social media users to develop and maintain close relationships with the influencers, such as with a good friendship. Accordingly, this study formulates the following research hypothesis:

***H1-1:*** 
*Social media users’ perception of language similarity in communication with their favorite influencer is positively associated with perceived friendship with the influencer.*


***H1-2:*** 
*Social media users’ perception of interest similar to their favorite influencer is positively associated with perceived friendship with the influencer.*


***H1-3:*** 
*Social media users’ perception of interaction frequency with their favorite influencer is positively associated with perceived friendship with the influencer.*


***H1-4:*** 
*Social media users’ perception of self-disclosure of influencer-related information is positively associated with perceived friendship with the influencer.*


One of the most powerful drivers of social media users’ digital content consumption behavior is to bring them pleasure, happiness, and enjoyment [6,7]. For example, if the digital content created by social media influencers is not able to fulfill social media users’ hedonic motivation, it may generate negative influences on users’ emotional and mental states, such as unhappiness [3]. In addition, social media users are likely to enjoy interacting with their favorite influencers on social media platforms and consuming the influencers’ digital content to escape from everyday life and/or relieve boredom [38,39]. The virtual interactions with social media influencers lead users to feel like they spend time with close friends with the same communication style and interests in the virtual world [40]. Additionally, social media platforms enable users to easily communicate with their favorite influencers and consume the influencers’ digital content regarding personal facts and stories [9]. Discussing influencers’ personal facts and stories leads social media users to feel happy as they perceive the influencers as close friends, contributing to their psychological well-being in the real world [3,40]. Based on that, the current research proposes the following hypothesis:

***H2-1:*** 
*Social media users’ perception of language similarity in communication with their favorite influencer is positively associated with psychological well-being.*


***H2-2:*** 
*Social media users’ perception of interest similar to their favorite influencer is positively associated with psychological well-being.*


***H2-3:*** 
*Social media users’ perception of interaction frequency with their favorite influencer is positively associated with psychological well-being.*


***H2-4:*** 
*Social media users’ perception of self-disclosure of influencer-related information is positively associated with psychological well-being.*


In general, good friendships between individuals result in each party’s psychological well-being as well as life satisfaction [3]. From a psychological perspective, the theory of human flourishing assumes that individuals attempt to build and maintain close relationships with others by continuously interacting with them in order to enhance their psychological well-being [41]. In other words, social media users are likely to enjoy life satisfaction and feel happier in the real world when perceiving that they succeed in developing and maintaining close relationships with their favorite influencers even though the relationship exists only in the virtual world [38,42]. Accordingly, social media users are more likely to interact with their favorite influencers in the virtual world to develop perceived close relationships or friendships with the influencers, leading users to emotionally reduce work and/or life stress [38,39]. Social media users’ perceived friendship with their favorite influencers established by virtual interactions on social media serves as a significant driver of developing a feeling of psychological well-being in their daily life [3]. Based on this notion, the following research hypothesis is established:

***H3-1:*** 
*Social media users’ perceived friendship with their favorite influencer is positively associated with psychological well-being.*


In the consumer behavior fields, consumers’ perceived friendships with a particular brand or company become a significant driver of their reciprocity, re-patronage, and loyalty toward the brand [43,44]. This is primarily because the relationship between a consumer and a brand leads the consumer to devalue other brands and forgive the brand’s service failure as the consumer perceives and deals with the brand as a close friend [9,45]. As a result, the consumer is more likely to be behaviorally loyal toward the brand by maintaining a good friendship with it via continuously purchasing and using its products/services [9,43,45]. In addition, individuals’ perceived friendships with an object could be expressed as their levels of passion about, trust or dependence toward, commitment to, and self-connection with the object, resulting in favorable behaviors toward the object [3,44]. Accordingly, this study proposes that social media users can build a close relationship with their favorite influencers via aggregated virtual interactions on social media into overall loyalty toward the influencers and products/services advertised and promoted by the influencers on social media. Therefore, this study formulates the following research hypothesis:

***H3-2:*** 
*Social media users’ perceived friendship with their favorite influencer is positively associated with loyalty toward the influencer.*


In general, individuals are likely to form a sense of happiness when remembering and/or experiencing a particular object, environment, or situation. Hence, good memories and positive experiences with a specific environment, object, or situation lead individuals to formulate a sense of happiness and make individuals more optimistic about the environment, object, or situation [46]. In other words, social media users with good memories and positive experiences with their favorite influencers are more likely to behave favorably for the influencers, expecting higher levels of good memories and positive experiences, core components of happiness in their daily life (i.e., psychological well-being in this study) [3,47]. For example, social media users are more likely to support their favorite influencers who can bring users happiness and help users to maintain positive emotional states in the virtual world. The virtually formulated happiness and positive emotional states among social media users can be transferred to their psychological well-being in the real world [47]. Hence, social media users are more likely to purchase products/services advertised and promoted by their favorite influencers to enhance their level of psychological well-being formed by the virtual interactions with the influencers [3,46]. Accordingly, this study proposes the following research hypothesis:

***H4:*** 
*Social media users’ psychological well-being is positively associated with loyalty toward their favorite influencer.*


As a phenomenon, the spillover effect means that “a message influences beliefs related to attributes that are not contained in the message” [48] (p. 458). The spillover effect has been studied within various contexts, such as different brands within the same firm, different attributes of a company and/or brand, and between a firm and a nonprofit organization in partnership with the firm in cause-related marketing situations [49,50]. Particularly, in the socially responsible communication context, a spillover effect may occur between a brand and its spokesperson or ambassador. According to the associative network theory of memory, individuals’ memory of a particular object (or product) tends to be formed as a web or network of interconnected conceptual nodes in their brains that represents a variety of different pieces of information about the object (or product) with varied associative strength [51,52]. Therefore, individuals store the different pieces of information about the product in memory as conceptual nodes that are complicated and interconnected with other associative links, such as the products’ spokesperson and brands [53]. Since individuals’ product nodes have already been interconnected with the spokesperson and brand nodes within the same network, the product nodes will be influenced by external factors if the spokesperson and brand nodes are influenced by those external factors. Based on the fundamental notion of the associative network theory of memory, the current research predicts the spillover effect of social media influencers’ (spokesperson) perceived social responsibility on users’ behavioral intention to purchase products advertised by the influencers on social media. In other words, this study assumes that social media users are more likely to positively evaluate and purchase the products their favorite influencers advertise and promote on social media when perceiving the influencers as more socially responsible, and vice versa. Thus, the following research hypothesis is formulated:

***H5:*** 
*Social media users’ perceived social responsibility of their favorite influencer moderates the relationships between their perceived friendship/psychological well-being and loyalty toward the influencer.*


## 3. Method

### 3.1. Data Collection

In this study, the unit of analysis was social media users in the United States subscribing to their favorite social media influencers’ pages via YouTube or following the influencers via Facebook, Instagram, or Twitter. These social media platforms were specifically chosen because a majority of Americans have commonly used YouTube (81%), Facebook (69%), Instagram (40%), and Twitter (23%) according to the “Survey of U.S. adults conducted from 25 January to 8 February 2021.” The first page of the questionnaire addressed a brief description of a social media influencer to help the participants to understand the study context. To collect data from representative samples, the second page of the questionnaire provided the participants with two screening questions carefully reviewed by the authors: (1) Who is your favorite social media influencer? (i.e., the participants’ responses to this question were carefully checked by the authors by searching for the addressed names of social media influencers on Google; based on the social media influencers-related information on Google, the authors could double-check whether the influencers addressed by the participants were playing a role as social media influencers conceptually well-aligned with the definition of social media influencers in this study); and (2) Please write two main reasons why you like the social media influencer (the participants with simple answers, such as “I don’t know” or “None,” were removed during the authors’ data purification process). The second question was also designed to arouse the participants’ cognition of their addressed social media influencer before responding to the next questions. The authors posted the online survey link on Amazon’s Mechanical Turk in the second week of May 2021 until the recommended number of participants was met for multivariate analyses to confirm reliabilities/validities (i.e., confirmatory factor analysis) and test the hypothesized associations among the variables in this study (i.e., structural equation modeling) (*N* = 388). The demographic characteristics of the participants in this study used for data analyses are indicated in Table 1.

### 3.2. Measures

The authors adapted and revised multiple items to measure language similarity, interest similarity, interaction frequency, self-disclosure, perceived friendship, psychological well-being, loyalty, and perceived social responsibility of social media influencers. The authors invited two professionals to review the revised items and operationalizations of all constructs before conducting a pilot test. Next, the authors conducted a pilot test with 50 undergraduates at a public university in the United States, resulting in minor changes in the wording of some items and overall flow of the questionnaire, and then finalized the questionnaire. The participants were asked to respond to all questions with a 7-point Likert scale anchored by “1 = strongly disagree” and “7 = strongly agree” except for demographic characteristics. This study referred to the works of Su et al. [9] and Kim and Kim [3] to measure social media influencer attributes, consisting of language similarity (four items), interest similarity (three items), interaction frequency (four items), and self-disclosure (five items). Second, this study measured social media users’ perceived friendship with seven items from the work of Yim, Tse, and Chan [54]. Third, to measure psychological well-being, four items were adapted and revised from the work of Lee [55]. Fourth, the perceived social responsibility of influencers was measured with three items from Liu, Wong, Rongwei, and Tseng [56] (e.g., (1) “My favorite social media influencer supports nonprofit organizations working in socially problematic areas”; (2) “My favorite social media influencer contributes to campaigns and projects that promote the well-being of the society”; and (3) “My favorite social media influencer attempts to create a better life for others in need”). Lastly, this study operationalized loyalty with four items based on the work of Kim and Kim [57]. To avoid any issue with common method bias, the authors randomly ordered the survey items by employing a procedural remedy suggested by Podsakoff, MacKenzie, and Podsakoff [58].

## 4. Empirical Findings

### 4.1. Tests of Reliabilities and Validities

To test the reliabilities and validities of all measures, the authors used the two-step approach proposed by Anderson and Gerbing [59]. before empirically testing the research hypotheses through structural equation modeling. As the first step, with SPSS 28.0, the authors conducted reliability analysis by estimating Cronbach alpha coefficients of all variables measured with multiple items. As indicated in Table 2, all variables’ alpha coefficients were greater than the value of 0.70, which is generally acceptable in the social science fields [60].

As the second step, with AMOS 28.0, the authors performed a confirmatory factor analysis to rigorously test the validities of all measures. To maintain the level of convergent validity, one of the items with less than 0.50 of the standardized regression weight, measuring the perceived friendship construct, was removed. The estimated fit indices of the measurement model were overall acceptable to proceed with tests of validities: χ^2^ = 859.075, degree of freedom = 353 (the normed χ^2^ = 2.434), *p* < 0.001, RMSEA = 0.061, IFI = 0.915, TLI = 0.901, and CFI = 0.914 [61]. Table 2 indicates that all standardized regression weights and critical ratios of all items exceeded 0.50 and were statistically significant (*p* < 0.001), respectively, confirming the convergent validities of all measures. In addition, the authors estimated the composite construct reliabilities of all constructs based on the empirical findings of the confirmatory factor analysis, signifying convergent validities because the values were greater than 0.70 [61] (see Table 3).

The authors tested the discriminant validities of all constructs according to the recommendation of Fornell and Larcker [62]. First, the respective average variance extracted values were estimated based on the standardized regression weights of all items of each construct (see Table 3). Second, the authors conducted correlation analyses and compared the squared correlation coefficients of two constructs to the average extracted variance extracted values of each construct. In this case, the squared correlation coefficients of two constructs should not exceed the respective average variance extracted values of each construct to signify the discriminant validities of the two constructs. As demonstrated in Table 3, the average variance extracted values of all constructs were greater than the squared correlation coefficients, confirming the discriminant validities of all constructs used in this study.

Lastly, the authors conducted Harman’s one-factor test as a statistical remedy to check whether the procedural remedy used during the questionnaire development process controlled common method bias [58]. The measurement model’s χ^2^ and degree of freedom values were 859.075 and 353 (the normed χ^2^ = 2.434), whereas a one-factor model’s χ^2^ and degree of freedom values were 2366.695 and 377 (the normed χ^2^ = 6.278), respectively. Because the normed χ^2^ value of the measurement model was significantly better than that of the one-factor model (i.e., the measurement model has a higher level of factor-explanation power than that of the one-factor model), it was confirmed that the procedural remedy used in this study controlled common method bias well.

### 4.2. Tests of Research Hypotheses

With AMOS 28.0, the structural equation modeling approach was used to empirically test the research hypotheses based on maximum likelihood estimates of each path (see Figure 1). Compared to other multivariate techniques that investigate only a single relationship at a time, structural equation modeling enables scholars to examine the interrelationships among multiple independent variables, mediators, and dependent variables via one comprehensive technique, testing an entire theory with all possible variables [61]. Hence, the structural equation modeling approach was used by the authors because it would help test the key theoretical associations in the parasocial interaction, the self-congruity, and the psychological well-being theories through one comprehensive technique. The fit indices of the proposed model were acceptable to interpret the empirical findings in the social science fields: χ^2^ = 887.539, degree of freedom = 357 (the normed χ^2^ = 2.486), *p* < 0.001, RMSEA = 0.063, IFI = 0.911, TLI = 0.898, and CFI = 0.910 [61].

First, H1 speculated that social media users’ perceptions of digital attributes of their favorite influencer have positive influences on perceived friendship with the influencer. The empirical findings addressed that users’ perceived friendship with their social media influencer was significantly affected by their perceptions of language similarity in communication with the influencer (standardized regression weight = 0.464, critical ratio = 5.758, *p* < 0.01), interest similar to the influencer (standardized regression weight = 0.298, critical ratio = 2.332, *p* < 0.05), and self-disclosure of influencer-related information (standardized regression weight = 0.320, critical ratio = 3.191, *p* < 0.01), but not by interaction frequency (standardized regression weight = −0.182, critical ratio = −1.448, n.s.), supporting H1-1, H1-2, and H1-4. Second, H2 addressed that social media users’ perceptions of digital attributes of their favorite influencer have positive effects on psychological well-being. The empirical results indicated that users’ psychological well-being was not significantly influenced by their perceptions of language similarity in communication with the influencer (standardized regression weight = −0.039, critical ratio = −0.397, n.s.), interest similar to the influencer (standardized regression weight = 0.210, critical ratio = 1.404, n.s.), interaction frequency with the influencer (standardized regression weight = −0.046, critical ratio = −0.315, n.s.), and influencer-related information (standardized regression weight = 0.224, critical ratio = 1.896, n.s.), not supporting H2-1, H2-2, H2-3, and H2-4. From a statistical perspective, the possible reason for the insignificant paths from social media influencer attributes to psychological well-being was the relative powers between paths. Specifically, the empirical results of the correlation analysis indicated significant associations between four social media influencer attributes and psychological well-being (i.e., 0.309 0.463, 0.390, and 0.477, *p* < 0.01). However, the respective impacts of social media influencer attributes on perceived friendship were much stronger than the respective influences of social media influencer attributes on psychological well-being, making the paths from social media influencer attributes to psychological well-being weak and insignificant [61]. Instead, social media influencer attributes can increase the level of social media users’ psychological well-being only via perceived friendship due to the relative power between the social media influencer attributes and perceived friendship/psychological well-being constructs (see Table 4). Third, H3 is related to the positive associations between perceived friendship and psychological well-being/loyalty. Users’ perceived friendship with their favorite social media influencer had significant effects on their psychological well-being (standardized regression weight = 0.354, critical ratio = 3.471, *p* < 0.01) and loyalty toward the influencer (standardized regression weight = 0.465, critical ratio = 6.640, *p* < 0.01), supporting H3-1 and H3-2. Lastly, H4 is associated with the positive relationship between psychological well-being and loyalty. Users’ psychological well-being had a significant impact on loyalty toward their favorite social media influencer (standardized regression weight = 0.422, critical ratio = 6.219, *p* < 0.01), supporting H4 (see Table 4).

To test H5 (the moderating role of the perceived social responsibility of influencers in the relationship between users’ perceived friendship with the influencer/psychological well-being and loyalty toward the influencer), this study used a chi-squared difference test via a multigroup analysis with AMOS 28.0 (i.e., highly perceived social responsibility of influencer group (*N* = 203) vs. lowly perceived social responsibility of influencer group (*N* = 178) based on the standard deviation (SD) split approach (−1 *SD* = low group and +1 *SD* = high group)) [63,64]. The assumption of this approach is for “the denominated path to be equal across groups while the other, more general model allows this parameter to differ across groups” [65] (p. 946). Thus, the denominated paths of two groups (i.e., high and low) were compared to the non-restricted corresponsive paths of two groups (i.e., high and low), estimating significant changes in the respective χ^2^ values and degree of freedom values. The empirical findings found a significant difference in the path from users’ psychological well-being to loyalty toward their favorite social media influencer (i.e., unconstrained model: χ^2^ = 1294.862 and degree of freedom = 714 vs. constrained model: χ^2^ = 1298.335 and degree of freedom = 715; χ^2^ difference = 3.473, *p* < 0.10): low group (standardized regression weight = 0.353, critical ratio = 3.772, *p* < 0.01) vs. high group (standardized regression weight = 0.549, critical ratio = 5.189, *p* < 0.01), partially supporting H5. This empirical result could be interpreted that the influence of users’ psychological well-being on loyalty toward their favorite social media influencer would be stronger when users perceive the influencer as being more socially responsible.

## 5. Discussion

As one of the theoretical contributions to the existing literature on psychological well-being, this study expands the parasocial interaction theory by considering social media users’ psychological well-being as a new outcome of the digital attributes of social media influencers rather than influencer-focused variables, such as trust toward and identification with the influencer [24,32,57,66,67]. In other words, prior research in this field focused primarily on the perspectives of influencer-focused or brand-focused benefits [34,66,67]. The approach of previous empirical research did not consider social media users’ basic motivations for consuming digital content, such as enhanced quality of life and psychological well-being in the real world [6,7,30,40]. However, this study revealed that the parasocial interaction between social media users and influencers in the virtual world (or on social media platforms) could be transferred to users’ own psychological benefits, including perceived friendship and psychological well-being in the real world. In other words, the empirical results of this study addressed that the attributes of social media influencers could lead users to perceive that they have good friendships with the influencers although the attributes are digitalized on social media platforms, consequently resulting in the development of users’ psychological well-being [68,69]. As the technological features of social media platforms have developed, users will be more likely to actively interact with their favorite social media influencers, such as via the metaverse and virtual reality services, in order to be happier and more pleased in the real world [6,40]. Therefore, this study proposes a new research avenue that psychological well-being could be one of the significant outcomes of digitalized social media influencer attributes (i.e., language similarity, interest similarity, and self-disclosure indirectly via perceived friendship) and perceived friendship between social media users and influencers according to the fundamental notion of the parasocial interaction theory.

As the second theoretical contribution to the extant literature, this study considered the moderating role of users’ perceived social responsibility of social media influencers in the context of social media influencer marketing, compared to previous research that focused on users’ perceived social responsibilities of a brand or company that has a partnership with a social media influencer [13,68]. The approach of previous research on social media has overlooked the important role of influencers in advertising and promoting brands/products through influencers’ digital content on social media platforms [11,35,42]. The empirical findings of the current study found that users’ perceived social responsibility of their favorite social media influencers moderated the relationship between their psychological well-being and loyalty toward the influencers. In other words, as a digitalized spokesperson or a brand ambassador [7,50], the social responsibilities and values of social media influencers are as important as those of the brand and/or product. More seriously, prior research has indicated that social media influencers’ socially unacceptable scandals could generate the spillover effect on a brand and/or a product in partnership with the influencer [10,50]. Although previous empirical studies focused on social media influencers as a new digital marketing channel, they did not consider the socially responsible aspects of social media influencers mainly by emphasizing the socially and environmentally responsible aspects of a brand and a product in the consumer’s decision-making process [11,68]. By reflecting the ongoing issues with social media influencers’ socially unacceptable scandals, this study discovered that the path from users’ psychological well-being to loyalty toward their favorite social media influencers and products advertised by the influencers could be weakened if the users perceive the influencers as less socially responsible. Therefore, this study proposes new insights on social media influencer marketing and corporate social responsibility by emphasizing the important role of social media influencers’ social responsibilities and values in developing users’ loyalty toward the influencers and brands/products.

From a managerial perspective, social media influencers should enhance users’ perceived friendship with the influencers and psychological well-being, leading users to be more loyal toward the influencers and brands/products promoted by the influencers. Although companies have strategic partnerships with social media influencers as new marketing channels or invest in the production of the influencers’ digital content as a viral marketing strategy, companies and influencers need to think about how to develop good friendships with social media users and how to make them happy via their digital attributes, leading users to purchase products/services advertised by the influencers. Above all, however, users perceive their favorite social media influencers as close friends, resulting in higher levels of users’ psychological well-being and loyalty. The empirical results of the current research found that users’ perceived friendships with their favorite social media influencers were enhanced by higher levels of perceived language similarity, interest similarity, and self-disclosure.

First, social media influencers should actively use social media tools, such as comments, posts, and messages, to indicate their communication style. By clearly indicating their communication style, social media influencers may make users unconsciously follow their communication style, leading users to perceive a high level of language similarity with the influencers. In addition, social media influencers may be able to use voice communication tools to directly communicate with users and follow users’ communication styles during live streams or via digital content. This effort enables users to increase the level of their perceived language similarity with their favorite social media influencers, leading users to perceive the influencers as close friends in the real world. The formation of perceived friendship with social media influencers via digital interactions results in higher levels of users’ psychological well-being and increased loyalty toward the influencers and the products/services advertised and promoted by the influencers. Secondly, to enhance users’ perceived interest similarity with their favorite social media influencers, an influencer needs to work with a company that has a partnership with the influencer by collaboratively designing and providing a virtual community on social media platforms that share a list of the influencer’s current interests in the company’s products/services and talks with users about those interests and products/services. In addition to developing a virtual community, the influencer can provide live streaming services sponsored by the company to actively interact with users regarding feelings and opinions about the company’s interests and its products/services. This managerial recommendation means that influencers’ social media pages and live broadcasts can be used as a vehicle for delivering their interests and exchanging feelings and opinions about interests with users. This effort leads users to feel like their favorite social media influencers have similar interests to theirs and will respond to their opinions and feelings about the interests and products/services, enhancing their perception of friendship with the influencers. Third, the empirical findings of this study indicate that social media users’ perceived friendship would be enhanced by their favorite influencers’ openness about themselves to users (e.g., personal facts and feelings). This means that social media users perceive their favorite influencers as close friends by knowing more about the influencers via digital content consumption. Hence, social media influencers need to share their real feelings/emotions and personal facts with users, acting as real friends, which in turn fosters influencer-user closeness. Fourth, this study confirmed the significant moderating role of users’ perceived social responsibility of social media influencers in the relationship between users’ psychological well-being and loyalty. When users perceive their favorite social media influencers as more socially responsible, the impact of psychological well-being on loyalty is significantly stronger. This empirical finding implies that users’ evaluation of social responsibilities and values of social media influencers serves as an important component in the decision-making process (i.e., purchase behavior and positive word-of-mouth). Therefore, to strengthen the association between users’ psychological well-being and loyalty toward their favorite social media influencers and products/services, influencers and companies need to design and implement socially responsible campaigns. For example, social media influencers get support from companies for the production of “the finding and helping people in need content” that donates money or products/services of companies for them. In addition, social media influencers may donate some portion of the money to nonprofit organizations or campaigns if users purchase a company’s products/services with the names of influencers. In this case, users feel like they are working with their favorite social media influencers for society together, just like a win-win situation. These collaborative efforts of social media users, influencers, and companies in the virtual world can generate positive effects on society in the real world.

## 6. Conclusions

The main purpose of this study was to predict users’ loyalty toward their favorite social media influencers from the perspectives of perceived friendship and psychological well-being, which were initially determined by social media influencer attributes, including perceived language similarity, interest similarity, interaction frequency, and self-disclosure. More importantly, this study found the significant moderating role of users’ perceived social responsibility of their favorite social media influencers in the relationship between psychological well-being and loyalty toward the influencer. The empirical findings of this study proposed a new insight into social media influencer marketing, particularly within the context of strategic partnerships between a brand/product and social media influencers.

## 7. Limitations

The first limitation of the current research is related to participants from different social media platforms. Specifically, this study collected data from social media users on various platforms, from Twitter (text-based social media) to YouTube (video-based social media). Each social media platform provides social media users with unique digital attributes which might have had different impacts on perceived friendship, psychological well-being, and loyalty toward their favorite influencers, depending on the social media platforms’ services. Although there were no significant differences in the mean values of all constructs between social media platforms in this study, there is the possibility that different digital services of social media platforms serve as an important moderator. Therefore, future research should employ a mixed-method approach by conducting multiple in-depth interviews or focus group interviews with social media users from different platforms in order to compare differences in digital attributes, friendship, psychological well-being, and loyalty.

The second limitation is associated with this study’s approach to the characteristics of social media influencers. This study focused primarily on the socially responsible aspects of social media influencers without consideration of other aspects, including types of influencers’ digital content (e.g., food, health, or beauty), demographic characteristics (e.g., gender, education level, age, or race), and physical and psychological characteristics (e.g., physical attractiveness, personal values, or personality traits) [69]. The construct of perceived social responsibility of social media influencers might have been interactively influenced by various aspects of the influencers, including backgrounds and past histories of ethical scandals. Therefore, future research needs to better control various aspects of social media influencers by revealing the pure effect of perceived social responsibility of influencers on users’ psychological well-being and loyalty. In addition, this study did not consider the moderating roles of social media users’ demographic characteristics in the proposed model because the participant composition was skewed toward female and younger users. Thus, this study conducted independent *t*-tests and one-way ANOVAs to find the significant differences between demographic groups (e.g., female users vs. male users, 20s users vs. 30s users, Facebook users vs. YouTube users vs. Instagram users, and large amounts of time on a social media group vs. small amounts of time on a social media group) after randomly selecting 50 samples from each group. Although there were no significant differences between demographic groups in this study, future research should consider the roles of social media users’ demographic characteristics along with their psychological characteristics, including personal values and personality traits [57].

The third limitation of the current research is based on the classification of social media influencer attributes. This research identified and classified social media influencer attributes such as language similarity, interest similarity, interaction frequency, and self-disclosure based on the existing literature on the parasocial interaction theory and social media. However, there is the possibility that digital attributes consist of more various sub-dimensions, depending on the subject and social media platforms. Therefore, future research should conceptually and empirically identify and classify sub-dimensions of digital attributes by conducting a comprehensive literature review in these fields, analyzing the comments and reviewing data regarding social media influencers.

## Figures and Tables

**Figure 1 ijerph-19-02362-f001:**
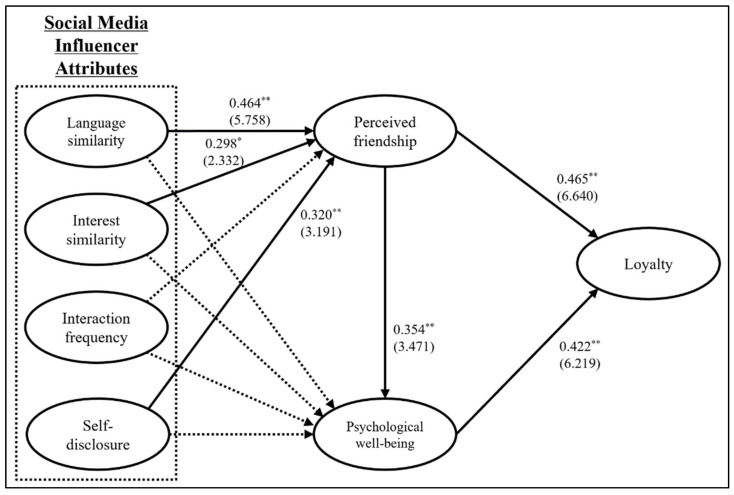
Estimates of structural equation modeling.** *p* < 0.01, * *p* < 0.05. Note. Standardized regression weight (critical ratio), solid line: significant path, dotted line: insignificant path.

**Table 1 ijerph-19-02362-t001:** Demographic analysis of respondents.

Demographic Variables		*N* = 381	Percent (%)
Gender	Female	265	69.6%
	Male	116	30.4%
Age	20s	267	70.1%
	30s	98	25.7%
	40s	16	4.2%
The most-used social media platform to interact with the favorite influencer	Facebook	103	20.7%
	Twitter	29	7.6%
	Instagram	101	26.5%
	YouTube	136	36.5%
	Others	12	2.4%
Education	High school	78	20.4%
	College or university degree	270	70.9%
	Graduate degree	33	8.7%
Vocation	Self-employed	65	17.1%
	Employed	96	25.2%
	Out of work and looking for work	33	8.7%
	Homemaker	60	15.7%
	Student	97	25.5%
	Military	21	5.5%
	Other	9	2.3%
The amount of time spent on social media per day	Less than 2 h	30	7.9%
	Between 2 and 4 h	190	49.9%
	More than 5 h	161	42.3%

**Table 2 ijerph-19-02362-t002:** Results of confirmatory factor analysis for items.

Constructs and Items	Standardized Regression Weight	Critical Ratio
Language similarity (α = 0.820)		
My favorite social media influencer uses a similar communication style (language, phrases, terms, etc.) as me on the social networking pages.	0.632	Fixed
My favorite social media influencer communicates in a way similar to how I talk with my friends on social networking pages.	0.577	14.921
There is no big difference between the communication styles used by me and my favorite social media influencer on social networking pages.	0.796	11.372
I like the communication style used by my favorite social media influencer on social networking pages.	0.720	10.802
Interest similarity (α = 0.762)		
I am interested in what my favorite social media influencer talks about on social networking pages.	0.821	Fixed
My favorite social media influencer shares similar interests as I do while communicating on social networking pages.	0.559	10.984
I like the content of messages posted by my favorite social media influencer on the social networking pages.	0.832	17.709
Interaction frequency (α = 0.793)		
How often do you view the messages (e.g., text, picture, or video) posted by your favorite social media influencer on social networking pages?	0.735	Fixed
How often do you click “like” for the messages posted by your favorite social media influencer on social networking pages?	0.784	14.335
How often do you comment on the messages posted by your favorite social media influencer on social networking pages?	0.612	11.216
How often do you share the messages posted by your favorite social media influencer on social networking pages?	0.738	13.522
Self-disclosure (α = 0.814)		
My favorite social media influencer voluntarily shares personal facts with followers on social networking pages.	0.621	Fixed
My favorite social media influencer is open about his or her feelings to followers on social networking pages.	0.557	11.150
My favorite social media influencer is quite open about his/herself to followers on social networking pages.	0.819	12.267
My favorite social media influencer reveals a lot of facts about his/herself on social networking pages.	0.622	10.084
My favorite social media influencer says much about his/herself on social networking pages.	0.785	11.954
Psychological well-being (α = 0.823)		
My favorite social media influencer’s pages satisfy my overall needs.	0.813	Fixed
My favorite social media influencer’s pages play a very important role in my social well-being.	0.848	16.668
My favorite social media influencer’s pages play a very important role in my leisure well-being.	0.709	14.073
Loyalty (α = 0.768)		
I intend to purchase products associated with my favorite social media influencer.	0.643	Fixed
I make an effort to search for my shopping needs related to my favorite social media influencer.	0.542	9.117
I encourage friends and relatives to shop for products related to my favorite social media influencer.	0.819	12.527
I say positive things about my favorite social media influencer to others.	0.780	12.170

**Table 3 ijerph-19-02362-t003:** Construct intercorrelations (*Φ*), mean, SD (standard deviation), AVE (average variance extracted), and CCR (composite construct reliability).

	1	2	3	4	5	6	7
1. Language similarity	1						
2. Interest similarity	0.495 **	1					
3. Interaction frequency	0.542 **	0.682 **	1				
4. Self-disclosure	0.434 **	0.620 **	0.632 **	1			
5. Perceived friendship	0.526 **	0.569 **	0.501 **	0.582 **	1		
6. Psychological well-being	0.309 **	0.463 **	0.390 **	0.477 **	0.507 **	1	
7. Loyalty	0.347 **	0.491 **	0.437 **	0.533 **	0.591 **	0.564 **	1
Mean	4.683	5.782	5.550	5.701	5.330	5.549	5.808
SD	1.327	1.059	1.040	1.012	1.246	1.318	1.160
AVE	0.471	0.560	0.519	0.474	0.536	0.628	0.497
CCR	0.778	0.787	0.810	0.815	0.873	0.834	0.794

* *p* < 0.05, ** *p* < 0.01.

**Table 4 ijerph-19-02362-t004:** Standardized structural estimates.

	Path	Standardized Regression Weight	Standardized Error	Critical Ratio
H1-1	Language similarity→Perceived friendship	0.464	0.086	5.758 **
H1-2	Interest similarity→Perceived friendship	0.298	0.149	2.332 *
H1-3	Interaction frequency→Perceived friendship	−0.182	0.168	−1.448
H1-4	Self-disclosure→Perceived friendship	0.320	0.119	3.191 **
H2-1	Language similarity→Psychological well-being	−0.039	0.104	−0.397
H2-2	Interest similarity→Psychological well-being	0.210	0.172	1.404
H2-3	Interaction frequency→Psychological well-being	−0.046	0.191	−0.315
H2-4	Self-disclosure→Psychological well-being	0.224	0.138	1.896
H3-1	Perceived friendship→Psychological well-being	0.354	0.101	3.471 **
H3-2	Perceived friendship→Loyalty	0.465	0.049	6.640 **
H4	Psychological well-being→Loyalty	0.422	0.048	6.219 **
	**Indirect Path**	**Standardized Indirect** **Effects**	**95%** **Bootstrapping Confidence** **Intervals**	***p*-Value**
	Language similarity→Psychological well-being	0.165	0.063~0.292	0.004
	Interest similarity→Psychological well-being	0.106	0.014~0.259	0.020
	Interaction frequency→Psychological well-being	−0.065	−0.229~0.013	0.085
	Self-disclosure→Psychological well-being	0.114	0.032~0.230	0.033
	**Endogenous Variables**	**SMC (R^2^)**		
	Perceived friendship	0.642		
	Psychological well-being	0.421		
	Loyalty	0.629		

* *p* < 0.05, ** *p* < 0.01.

## Data Availability

The data presented in this study are available on request from the corresponding author. The data are not publicly available due to privacy reasons.

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
