# Peer review of "Rise of Social Media Influencers as a New Marketing Channel: Focusing on the Roles of Psychological Well-Being and Perceived Social Responsibility among Consumers"

_ijerph, 2022, doi:10.3390/ijerph19042362_

Round 1

Reviewer 1 Report

Fascinating look at parasocial relationships and how social media influencer responsibilities and offline behaviors impact user/consumer interactions. This has strong links with existing corporate behavior, so it makes sense to tie in with social media influencers and creators, especially on YouTube. 

How does the study take into account the amount of time users spend on social media? Does it also look at various other users? For example, how would subscribers to "BreadTube" content creators vary from others potentially taking the survey? A follow-up may inquire into how specific user personae are linked to specific behaviors in commenting and parasocial relationships. 

Great model and study setup. Excellent writeup, please check for some instances of plural v. singular usage. 

Author Response

Please refer to the attached response letter.

Reviewer 2 Report

  1. page 2, line 45 here you can insert a citation that shows a gap or need to investigate these specific attributes that relate to loyalty.
  2. line 80. this study's argument... this sentence need to be rephrased to align with academic writing.
  3. section 2.1. and section 2.2 It will be helpful for the reader to introduce what these theories talk about and what attributes do they consider. Provide a reasoning when additional attributes or factors are added for the sake of the study.
  4. The method, for what reason these platforms specifically were chosen? Support your choice.
  5. How did you select the influencers on Google? based on what, kindly support your choice
  6.  Age for instance can be one of indicators also to consider cross platforms. Any reflection regarding the demographics?
  7. Line 423, how does structural equation modeling here helps in achieving the purpose. for what reason this analysis fits best your purpose?

Author Response

(The authors gave the same response as above.)

Reviewer 3 Report

Overall, this an interesting paper. The lit review is ok, the theory selection is appropriate, the methodology is solid and well explained, and the paper has good organization and flow. Suggestions for improving the text are below.

Written quality

  1. The written quality is ok but there are some typos / grammatical mistakes and awkward word choices. It doesn’t need professional editing, just ask a friend proof read the text, checking the choice of articles (the, a, an, or), pluralization, and word choice.
  2. Also some unnecessary wordiness, “Accordingly, based on the parasocial interaction theory, this study proposes that social media users tend to be able to develop perceived friendships with 58 their favorite influencers in…” (pg 2) could be “Accordingly, we draw from the parasocial interaction, the self-congruity, and the psychological well-being theories to propose that social media users develop perceived friendships with their favorite influencers in…” Note how this revised sentence introduces all of the theories used in the paper early on instead of surprising the reader by introducing these last two theories in the lit review.
  3. The word “context” is used unnecessarily in several places, such as on page 1, line 42 “in the traditional advertising context” could just be “in traditional advertisements.” Can alo remove “context” from page 4, line 197.

Introduction

  1. The study’s motivation (why you are studying this phenomenon) needs to be stronger. This sentence is too much of a leap: “Therefore, 45 there is a need to empirically investigate what drives social media users’ loyalty toward their favorite influencers, such as repeat purchase behavior, positive word-of-mouth, and recommendation of products advertised by social media influencers [5].” Adding a brief discussion of how SM users’ loyalty leads to repeat purchase behavior, positive word-of-mouth, and recommendation of products would help. In other words, why would I, the reader, care about loyalty to social media influencers? There are a couple of sentences on page 4 (lines 187-190) that might help build a better motivation for the study.
  2. The paragraph on page 2 about social media influencers social responsibility is not well integrated into the introduction…it feels just stuck into the intro…
  3. Also, there is an incomplete sentence in the introduction following the first citation. And wouldn’t say that “social media is becoming an important part of younger generation’s lives.” We’re a bit past that…Facebook has been heavily used for over 15 years but isn’t used much by Gen Z. I would just revise this sentence to say, “Social media use is widespread and has been shown to play an important role in the lives of both adolescents and young adults.” Or some such.

Lit review: There should be some mention of the deleterious effects of social media use in the psychological well-being section (2.4) on page 4.

 Discussion

  1. Add to discussion potential explanations for why well-being was not found to be influenced by social media influencer attributes. One explanation could be social comparison theory, in particular upward social comparison, but there could be other/ better possible explanations in the literature on the dark side of social media. This additional doesn’t have to be lengthy and the current discussion’s text could be streamlined to make room for it.
  2. The following sentence starting on line 495 is a bit confusing: “Therefore, this study proposes a new research avenue that psychological well-being could be one of the significant outcomes of digitalized social media influencer attributes and perceived friendship between social media users and influencers according to the fundamental notion of the parasocial interaction theory” This reads as though the study found a direct positive association between influencer attributes and psych. well-being but these hypotheses were not supported.

Author Response

(The authors gave the same response as above.)

Reviewer 4 Report

This study conceptually identified social media influencer attributes as language similarity, interest similarity, interaction frequency, and self-disclosure, and examined the respective effects of each dimension on perceived friendship and psychological well-being, consequently resulting in loyalty toward social media influencers. I offer a few suggestions for modification.

Originality:  
Does the paper contain new and significant information adequate to justify publication?

The social media influencer attributes, perceived friendship, psychological well-being, loyalty, and social responsibility mentioned in this article have many similar discussions in related fields, and it is difficult to find the innovation of this paper.

Methodology:  
Is the paper's argument built on an appropriate base of theory, concepts, or other ideas?  Has the research or equivalent intellectual work on which the paper is based been well designed?  Are the methods employed appropriate?

>It is very far-fetched to use questions to summarize each dimension in the questionnaire, such as Psychological well-being (satisfy my overall needs, social well-being and leisure well-being), Loyalty ( purchase products, search for my shopping needs, encourage friends and relatives to shop products).

>In addition, the research variables are too complicated. Besides discussing the impact of social responsibility, why did the author also include the issue of business conduct?

Implications for research, practice and/or society:  
Does the paper identify clearly any implications for research, practice and/or society?  Does the paper bridge the gap between theory and practice? 

>The research results are slightly confusing and readers cannot intuitively connect with the research purpose of the paper.

>For Discussion and Conclusions, they have too little related discussion with literature or related theory.

>I suggest that the author should consider the research contribution and purpose of the paper, delete unnecessary content and paragraphs, and rewrite the manuscript.

Author Response

(The authors gave the same response as above.)

Round 2

Reviewer 4 Report

The authors have revised all review suggestions.